# TrueTeacher: Learning Factual Consistency Evaluation with Large Language Models

**Zorik Gekhman**[T,G,*]    **Jonathan Herzig**[G]    **Roee Aharoni**[G]
**Chen Elkind**[G]    **Idan Szpektor**[G]
[T]Technion - Israel Institute of Technology    [G]Google Research
zorik@campus.technion.ac.il
{zorik|jherzig|roeeaharoni|chenel|szpektor}@google.com

## Abstract

Factual consistency evaluation is often conducted using Natural Language Inference (NLI) models, yet these models exhibit limited success in evaluating summaries. Previous work improved such models with synthetic training data. However, the data is typically based on perturbed *human-written* summaries, which often differ in their characteristics from real *model-generated* summaries and have limited coverage of possible factual errors. Alternatively, large language models (LLMs) have recently shown promising results in directly evaluating generative tasks, but are too computationally expensive for practical use. Motivated by these limitations, we introduce TrueTeacher, a method for generating synthetic data by annotating diverse *model-generated* summaries using a LLM. Unlike prior work, TrueTeacher does not rely on human-written summaries, and is multilingual by nature. Experiments on the TRUE benchmark show that a student model trained using our data, substantially outperforms both the state-of-the-art model with similar capacity, and the LLM teacher. In a systematic study, we compare TrueTeacher to existing synthetic data generation methods and demonstrate its superiority and robustness to domain-shift. We also show that our method generalizes to multilingual scenarios. Lastly, we release our large-scale synthetic dataset (1.4M examples), generated using TrueTeacher, and a checkpoint trained on this data.[1]

## 1 Introduction

Generative summarization models are prone to generate summaries that are factually inconsistent with respect to the corresponding input documents (Goodrich et al., 2019; Kryscinski et al., 2019), limiting their applicability in real-world scenarios.

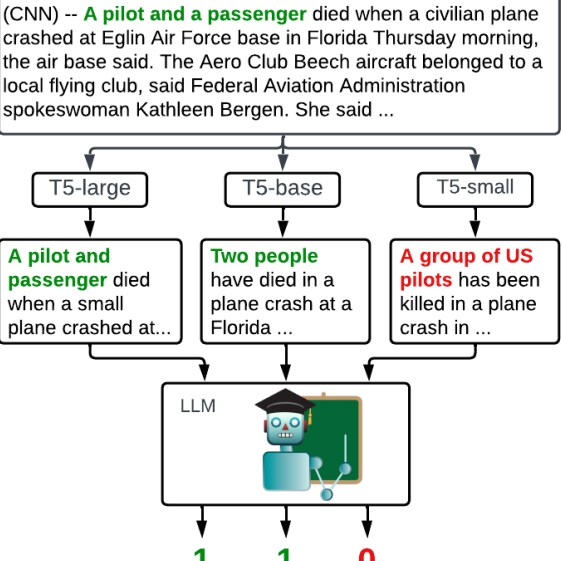

Figure 1: A real example from our data generation process. We fine-tune summarization models with different capacities, and use them to produce a diverse set of *model-generated* summaries of CNN/DM articles, which we label for consistency using a 540B LLM.

Since factual consistency evaluation could be cast as a Natural Language Inference (NLI) task, NLI models are often used to evaluate consistency (Falke et al., 2019a; Maynez et al., 2020; Laban et al., 2022). However, NLI models exhibit limited success in evaluating factual consistency in *summarization* (Falke et al., 2019b; Kryscinski et al., 2020), since NLI datasets lack the entailment phenomena that naturally arise in abstractive summarization (Khot et al., 2018). For example, single-sentence premise-hypothesis pairs are shorter than document-summary pairs (Mishra et al., 2021; Schuster et al., 2022).

To address this domain mismatch, previous work proposed various approaches for generating synthetic training data (Kryscinski et al., 2020; Yin et al., 2021; Utama et al., 2022; Balachandran et al., 2022). The data is typically generated by perturb-

---

*Work done during an internship at Google Research.

[1]Our dataset and model are available at:
https://github.com/google-research/
google-research/tree/master/true_teacher

ing human-written summaries to introduce factual inconsistencies. While these perturbations are effective, they are limited to factual error categories that can be covered by the perturbation logic. In addition, since simulating factual errors is challenging, such perturbations may fail to introduce factual errors, leading to incorrect labels.[2] Finally, since the synthetic summaries are based on *human-written* summaries, they may differ in style from real *model-generated* summaries, which can reduce the effectiveness of the synthetic data.

An alternative approach to augmenting NLI models with synthetic data, is to directly prompt large language models (LLMs) to evaluate factual consistency. Recently, there has been a growing evidence for the effectiveness of LLMs in evaluating generative tasks (Kocmi and Federmann, 2023; Wang et al., 2023; Liu et al., 2023), including factual consistency in summarization (Chen et al., 2023). However, LLMs are still too computationally expensive to be heavily used in practice.

To make the best of both worlds we propose TrueTeacher, a simple and effective synthetic data generation method that leverages *model-generated* summaries and the reasoning abilities of LLMs (Huang and Chang, 2022). In TrueTeacher, we first train a diverse collection of summarization models with different capacities. Next, we use these models to summarize each document in a given corpus (Figure 1). The resulting document-summary pairs are then annotated by prompting a LLM to predict the corresponding factual consistency label.

We apply TrueTeacher using FLAN-PaLM 540B (Chung et al., 2022) to generate a large-scale synthetic dataset, which is used to train a student model. Experiments on the summarization subset of the TRUE benchmark (Honovich et al., 2022) show that augmenting existing NLI data with TrueTeacher data improves a state-of-the-art model's ROC-AUC from 82.7 to 87.8, while maintaining similar model capacity. The resulting model even outperforms its LLM teacher, despite the latter having a $\times 50$ larger capacity.

We also compare TrueTeacher to existing synthetic data generation methods. To this end, we design a systematic study to re-evaluate existing methods with a "fair comparison" in a challenging setting. Our results indicate that existing approaches fail to generalize to documents derived from a distribution different from the one used for

synthetic data generation. In contrast, TrueTeacher demonstrates robustness by successfully generalizing to documents from new domains.

Finally, we apply TrueTeacher to generate *multilingual* synthetic data. While existing data generation methods are often limited to English (Utama et al., 2022; Balachandran et al., 2022), TrueTeacher can use a multilingual LLM. Results on the mFACE dataset (Aharoni et al., 2022), show improvements on 35 out of 45 languages when using our method. This demonstrates the usefulness of multilingual synthetic data and the effectiveness of TrueTeacher in generating such data.

To summarize, this work includes the following contributions:

- We introduce TrueTeacher, a synthetic data generation approach based on annotating *model-generated* summaries with LLMs, and demonstrate its effectiveness and robustness.

- We evaluate FLAN-PaLM 540B on the task of factual consistency evaluation and show that its knowledge can be distilled into a significantly smaller model using our method.

- We conduct a systematic study, re-evaluating existing synthetic data generation methods for the task in an apples-to-apples comparison and identify their limitations.

- We perform the first experiment in generating multilingual synthetic data for factual consistency, and demonstrate its usefulness.

- We release a large-scale dataset comprised of 1.4 million TrueTeacher examples, and verify its quality with human evaluation. We additionally release a state-of-the-art consistency evaluation model trained on this data.[1]

## 2 TrueTeacher

In this section we describe TrueTeacher, our approach for generating synthetic examples for the task of factual consistency evaluation in summarization. Our main motivation is to use factual inconsistencies that occur in real *model-generated* summaries, instead of relying on perturbed human-written summaries. To this end, we generate a diverse set of summaries using generative summarization models of different capacities, and leverage a LLM to label them for factual consistency. Some of the generated summaries are expected to contain factual errors, and we hypothesize

---

[2]As we also demonstrate in §4.3.

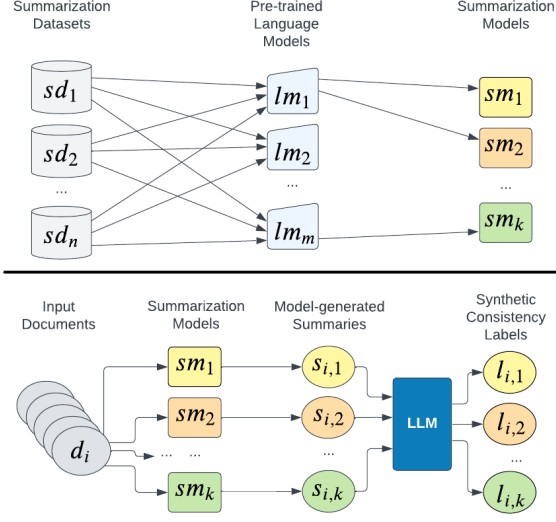

Figure 2: Our data generation process. We train a collection of generative summarization models, use them to summarize documents and label the resulting summaries for factual consistency using a LLM.

that a strong-performing LLM can generalize to the task and label them with sufficient quality to be useful for training. The usage of model-generated summaries not only yields more realistic texts, but also allows to potentially include rare errors, which can be harder to incorporate with perturbation logic.

Our data generation process is illustrated in Figure 2. First, we train a variety of summarization models (upper diagram). We use a collection of one or more summarization training sets $T = \{sd_1, sd_2, \ldots, sd_n\}$ and different pre-trained $LMs = \{lm_1, lm_2, \ldots, lm_m\}$ to fine-tune a collection of summarization models $SM = \{sm_1, sm_2, \ldots, sm_k\}$, where $k = n \times m$.[3] Using different pretrained LMs allows to diversify the expected consistency errors, e.g., errors made by large or small models. The choice of summarization training sets allows to control for the nature of the resulting summaries, e.g., focusing on abstractive training sets to increase output abstractiveness.

Next, we obtain *model-generated* summaries and annotate them (lower diagram). We choose a documents corpus $D = \{d_1, d_2, \ldots, d_r\}$ and use all the summarization models in $SM$ to summarize all the documents in $D$, resulting in a collection of model-generated output summaries $O = \{s_{1,1}, \ldots s_{r,k}\}$, where $s_{i,j}$ is the summary of document $d_i$ generated by summarization model $sm_j$. TrueTeacher

does not require gold summaries, which allows it to be used with any collection of documents $D$, and makes it more scalable than previous methods (Yin et al., 2021; Utama et al., 2022; Balachandran et al., 2022).

Finally, a LLM is prompted to label all the summaries in $O$ for consistency w.r.t. their source documents, resulting with labels $\{l_{1,1}, \ldots, l_{1,k}, \ldots l_{r,k}\}$.[4] Figure 1 illustrates a real example of this process for a single document $d_i \in D$. Each document, summary, and label $(d_i, s_{i,j}, l_{i,j})$ are then used as a synthetic example for training a factual consistency classifier. Since we leverage LLMs for labeling, our approach is likely to benefit from the ongoing progress in LLMs quality. Furthermore, previous approaches often rely on language-specific components (e.g., Information Extraction), which limits their applicability in multiple languages. Since recent LLMs are pretrained on multilingual data, our method can be easily applied to non-English languages, as we show in §5.

## 3 Experimental Setup

We use TrueTeacher to generate a synthetic dataset for factual consistency evaluation in summarization (§3.1), and experiment with it to evaluate the effectiveness and usefulness of our method (§4).

### 3.1 TrueTeacher Instantiation

To apply TrueTeacher, we instantiate the summarization datasets $T$, the pre-trained $LMs$ and the documents corpus $D$. We use XSum (Narayan et al., 2018) as $T$, T5 pre-trained models (Raffel et al., 2020) as $LMs = \{$T5-small, T5-base, T5-large, T5-3B, T5-11B$\}$, and documents from CNN/DailyMail (Hermann et al., 2015) as $D$.

As our teacher model, we employ FLAN-PaLM 540B (Chung et al., 2022). This model was instruction fine-tuned, including training on the closely-related NLI task.[5] Therefore, we expect it to generalize well to factual consistency evaluation.[6] We use zero-shot prompting for simplicity, and since applying few-shot or chain-of-thought prompting did not improve performance in early experiments.[7]

---

[3]We note that the pretrained $LMs$ here refer to the models that we are fine tuning for summarization, and they are different from the LLM that we use as the teacher.

[4]See §3.1 and §A.1 for our prompting implementation.

[5]https://github.com/google-research/FLAN/blob/e9e4ec6e2701182c7a91af176f705310da541277/flan/task_splits.py#L109

[6]We validate this expectation in §4.1 and §4.4.

[7]In §A.1 we discuss potential reasons to this.

| Summaries Source | # Consistent | # Inconsistent |
|---|---|---|
| T5-11B | 233,815 | 39,423 |
| T5-3B | 229,097 | 45,662 |
| T5-large | 195,681 | 81,986 |
| T5-base | 161,177 | 118,480 |
| T5-small | 88,129 | 190,012 |
| Total | 907,899 | 475,563 |

Table 1: Our generated dataset statistics.

Extensive implementation details about our FLAN-PaLM usage are provided in §A.1 and §A.2.

Applying TrueTeacher in this setup resulted in ~1.4M synthetic training examples (Table 1), which we use to train a student model for factual consistency evaluation.[8] In §4, we provide evidence for the dataset's quality through human evaluation (§4.4), its usefulness for improving NLI models in a challenging setting (§4.1), and its superiority over other existing synthetic datasets (§4.2).

In early experiments, we also explored data filtering based on prompting FLAN-PaLM for self-verification (details in §A.5). This resulted in an increase in the labeling accuracy. Yet, surprisingly, training the student model on the filtered data did not improve performance in comparison to training on the full dataset.[9] Thus, for simplicity, we conduct experiments using the full dataset.

### 3.2 Evaluation

To compare between consistency evaluation models, we use the TRUE benchmark (Honovich et al., 2022), focusing on its summarization subset: **MNBM** (Maynez et al., 2020), **FRANK** (Pagnoni et al., 2021), **SummEval** (Fabbri et al., 2020), **QAGS-X** and **QAGS-C** (Wang et al., 2020). For additional details about these datasets, we refer the reader to Honovich et al. (2022). Following Honovich et al., we use ROC-AUC in a binary classification setting as our evaluation metric.

### 3.3 Baselines

We compare the performance of factual consistency evaluation models trained on TrueTeacher data, against the top performing models on the TRUE benchmark: **QuestEval** (Scialom et al., 2021), $Q^2$ (Honovich et al., 2021), $\text{SUMMAC}_{ZS}$ (Laban et al., 2022), **T5-11B fine tuned on ANLI** (Honovich

et al., 2022), **WeCheck** (Wu et al., 2023), and the **Ensemble** from Honovich et al. (2022).[10]

We also compare TrueTeacher data generation *mechanism* to existing methods for synthetic data generation. We consider the following approaches:

**DocNLI** (Yin et al., 2021). Reformatted NLI, question answering and summarization datasets, including the CNN/DM corpus. The summarization-based positive examples are based on concatenated gold summaries. The negative examples are then generated using word/entity replacements.

**FactCC** (Kryscinski et al., 2020). The documents are from CNN/DM. The consistent summaries are randomly sampled sentences from the document, which are optionally injected with noise or paraphrased. The inconsistent summaries are obtained by rule-based transformations, such as sentence negation and entity/pronoun/number swaps.

**FactEdit** (Balachandran et al., 2022). The positive examples are based on gold summaries from CNN/DM. For the negative examples, an infilling model is trained using sentences from the documents, employing the OpenIE framework (Banko et al., 2007) to mask predicates and arguments. Each predicate and argument phrase in the summary is then iterativelly masked and infilled with the model's lower order beam candidates.

**Falsesum** (Utama et al., 2022). The positive examples are based on gold summaries from CNN/DM. For the negative examples, predicates and arguments are detected in the document and the summary using the OpenIE (Banko et al., 2007) framework. Randomly selected predicates and arguments from the summary are then masked and infilled using predicates and arguments from the document, or by "hallucinating" new content. For this purpose a dedicated infilling model is trained.

## 4 Experiments and Analysis

Our main experiments are in §4.1 and §4.2, followed by various analyses and ablations in §4.3, §4.4, §4.5 and §4.6. We design our experiments to address the following research questions (RQs):

- **RQ1:** What is the performance of FLAN-PaLM 540B in factual consistency evaluation in summarization? Is it a good choice for a teacher?

---

[8]Implementation details for our trained models are in §A.3.

[9]This could be attributed to the high-quality of the initial labels and the student model's robustness to noise.

[10]We discuss WeCheck in §6, and refer the reader to Honovich et al. (2022) for a detailed description of other baselines.

| | MNBM | QAGS-X | FRANK | SummEval | QAGS-C | Average |
|---|---|---|---|---|---|---|
| QuestEval (Scialom et al., 2021) | 65.3 | 56.3 | 84.0 | 70.1 | 64.2 | 68.0 |
| $Q^2$ (Honovich et al., 2021) | 68.7 | 70.9 | 87.8 | 78.8 | 83.5 | 77.9 |
| SUMMAC$_{ZS}$ (Laban et al., 2022) | 71.3 | 78.1 | 89.1 | 81.7 | 80.9 | 80.2 |
| T5-11B w. ANLI (Honovich et al., 2022) | 77.9 | 83.8 | 82.1 | 80.5 | 89.4 | 82.7 |
| WeCheck (Wu et al., 2023) | **83.0** | 81.4 | 88.1 | 79.8 | 82.6 | 83.0 |
| Ensemble (Honovich et al., 2022) | 76.6 | 85.8 | 91.2 | 82.9 | 87.7 | 84.8 |
| FLAN-PaLM 540B (Chung et al., 2022) | 76.0 | 88.1 | 91.4 | 83.7 | 85.2 | 84.9 |
| T5-11B w. ANLI + TrueTeacher full | 78.1 | **89.4** | **93.6** | **88.5** | **89.4** | **87.8** |

Table 2: ROC-AUC results on the summarization subset of the TRUE benchmark (Honovich et al., 2022).

- **RQ2:** Can TrueTeacher facilitate training of a competitive model w.r.t. state-of-the-art models?

- **RQ3:** What is the quality of the data generated using TrueTeacher compared to existing synthetic data generation methods?

We address RQ1 and RQ2 in §4.1. To address RQ1, we evaluate FLAN-PaLM 540B against competitive models for factual consistency evaluation. To address RQ2, we use our full dataset from §3.1 to train our best-performing model, and evaluate it in the exact same setting. Finally, RQ3 is addressed in §4.2, where we conduct a systematic study, comparing existing methods to TrueTeacher, while controlling for factors such as the synthetic data size and the documents used for data synthesis.

### 4.1 Main Results on the TRUE Benchmark

We address RQ1 by evaluating FLAN-PaLM 540B on the task and present the results in Table 2. FLAN-PaLM 540B achieves an impressive performance, with an average ROC-AUC of **84.9** compared to **83.0** of the best single-model baseline, and performs on-par with the Ensemble. This demonstrates the chosen LLM's capability for the task, and its potential as a teacher for smaller models.

To address RQ2, we fine-tune T5-11B (Raffel et al., 2020) over our full dataset (§3.1) mixed with ANLI (Nie et al., 2020). Table 2 shows that including TrueTeacher data in the training set, substantially improves the strong-performing **T5-11B w. ANLI** baseline from an average ROC-AUC of **82.7** to **87.8** (+5.1), while maintaining exactly the same model capacity. This strong result demonstrates the high effectiveness of TrueTeacher in a challenging setup. Notably, our model sets the new state-of-the-art result on the benchmark, outperforming the ×50 times larger LLM that we used as the teacher ($84.9 \rightarrow 87.8$). This can be

attributed to large-scale knowledge distillation on a specific task, while the LLM is trained to perform many tasks. Additionally, the smaller model is trained on target-domain data (documents and model-generated summaries) which can further improve performance (Gururangan et al., 2020).

### 4.2 Re-evaluating Synthetic Data Generation Methods – A Study

Previous studies on synthetic data generation have used different experimental setups, making it difficult to compare their results. In this section, we design a systematic study to re-evaluate existing methods in a standardized setup. We first discuss our study design choices followed by the results.

Previous work has demonstrated that synthetic data can improve NLI-based models. However, they typically used relatively small-capacity models, whereas Honovich et al. (2022) recently demonstrated significant performance gains by scaling up to T5-11B fine-tuned on ANLI. We therefore adopt this **competitive baseline**, to which we add synthetic data from each method. For ablation, we include variants trained solely on synthetic data (without ANLI), and also repeat our study using the smaller-capacity T5-base model.

To preform a fair comparison, we **restrict the number of examples** from each evaluated method to 100k, randomly sampled with balanced labels.

To evaluate **domain-shift robustness**, we further restrict the synthetic *training* examples to ones that were generated only based on CNN/DM documents,[11] and then consider the XSum-based evaluation sets as out-of-domain.[12]

---

[11] Some methods are based exclusively on CNN/DM while others use additional datasets, more details in §3.3.

[12] SummEval and QAGS-C are based on documents from CNN/DM, MNBM and QAGS-X use documents from XSum, and FRANK has documents from both CNN/DM and XSum. We split FRANK to FRANK-C and FRANK-X which contain its CNN/DN based and XSum based subsets respectively.

| Training data | CNN/DM-based | | | FRANK | XSUM-based | | | Average scores | | |
|---|---|---|---|---|---|---|---|---|---|---|
| | QAGS-C | SummEval | FRANK-C | FRANK | FRANK-X | QAGS-X | MNBM | In-domain | Out-of-domain | TRUE |
| **T5-11B** ANLI | 83.4 | 74.2 | 85.6 | 90.7 | 93.2 | **88.0** | 73.9 | 81.1 | 85.0 | 82.0 |
| FactEdit | 87.8 | 77.0 | 77.2 | 83.7 | 76.0 | 69.4 | 53.1 | 80.7 (-0.4) | 66.2 (-18.8) | 74.2 (-7.8) |
| FactEdit + ANLI | 88.9 | 78.9 | 81.1 | 88.0 | 86.1 | 76.2 | 59.8 | 83.0 (+1.9) | 74.0 (-11.0) | 78.4 (-1.6) |
| DocNLI | 89.1 | 72.9 | 83.0 | 89.2 | 92.4 | 83.8 | 67.0 | 81.7 (+0.6) | 81.1 (-3.9) | 80.4 (-1.6) |
| DocNLI + ANLI | 87.8 | 72.0 | 81.9 | 88.2 | 93.7 | 84.2 | 68.0 | 80.6 (-0.5) | 82.0 (-3.0) | 80.0 (-2.0) |
| FactCC | 83.1 | 79.0 | 81.6 | 84.1 | 67.5 | 72.7 | 55.0 | 81.2 (+0.1) | 65.1 (-19.9) | 74.8 (-7.2) |
| FactCC + ANLI | 84.7 | 83.3 | 84.7 | 89.5 | 89.6 | 82.9 | 71.5 | 84.2 (+3.1) | 81.3 (-3.7) | 82.4 (+0.4) |
| Falsesum | 90.3 | 85.4 | 85.8 | 89.8 | 84.5 | 70.8 | 53.9 | 87.2 (+6.1) | 69.7 (-15.3) | 78.0 (-4.0) |
| Falsesum + ANLI | **90.7** | 85.8 | 87.0 | 91.6 | 90.5 | 75.2 | 60.5 | **87.8** (+6.7) | 75.4 (-9.6) | 80.8 (-1.2) |
| TrueTeacher | 84.9 | 85.0 | 88.8 | 93.6 | **94.4** | 86.5 | 76.1 | 86.2 (+5.1) | 85.7 (+0.7) | 85.2 (+3.2) |
| TrueTeacher + ANLI | 88.4 | **85.8** | **89.6** | **93.9** | 93.9 | 87.8 | **76.3** | **87.9** (+6.8) | **86.0** (+1.0) | **86.4** (+6.4) |
| **T5-base** ANLI | 74.9 | 63.7 | 73.1 | 81.3 | 80.6 | 77.2 | 77.0 | 70.6 | 78.3 | 74.8 |
| FactEdit | 61.4 | 59.4 | 59.4 | 73.6 | 51.9 | 48.0 | 58.4 | 60.1 (-10.5) | 52.8 (-25.5) | 60.2 (-14.6) |
| FactEdit + ANLI | 68.7 | 60.0 | 62.2 | 78.5 | 73.6 | 72.2 | 75.5 | 63.6 (-7.0) | 73.8 (-4.5) | 71.0 (-3.8) |
| DocNLI | 71.4 | 66.5 | 66.7 | 77.9 | 81.0 | 75.2 | 71.6 | 68.2 (-2.4) | 75.9 (-2.4) | 72.5 (-2.3) |
| DocNLI + ANLI | 75.2 | 66.7 | 74.4 | 84.9 | 83.3 | 78.7 | 74.8 | 72.1 (+1.5) | 78.9 (+0.6) | 76.1 (+1.3) |
| FactCC | 74.0 | 72.7 | 78.7 | 83.2 | 71.9 | 71.0 | 62.7 | 75.3 (+4.7) | 68.5 (-9.8) | 72.7 (-2.1) |
| FactCC + ANLI | 72.8 | 73.2 | 78.8 | 83.2 | 66.8 | 71.5 | 63.2 | 74.9 (+4.3) | 67.2 (-11.1) | 72.8 (-2.0) |
| Falsesum | 80.9 | 74.2 | 82.0 | 86.4 | 71.6 | 65.0 | 53.1 | 79.0 (+8.4) | 63.2 (-15.1) | 71.9 (-2.9) |
| Falsesum + ANLI | **82.9** | 73.4 | **83.3** | 86.5 | 72.6 | 66.0 | 58.7 | 79.9 (+9.3) | 65.8 (-12.5) | 73.5 (-1.3) |
| TrueTeacher | 77.3 | 73.6 | 79.1 | 88.0 | 82.6 | 79.9 | 78.3 | 76.7 (+6.1) | 80.3 (+2.0) | 79.4 (+4.6) |
| TrueTeacher + ANLI | 81.9 | **78.0** | 81.4 | **89.3** | **86.4** | **81.9** | **78.5** | **80.4** (+9.8) | **82.3** (+4.0) | **81.9** (+7.1) |

Table 3: ROC-AUC results on TRUE comparing different synthetic data generation methods. For each model size, average scores are compared to the corresponding ANLI-only baseline (difference is listed in parentheses).

Table 3 presents the results of our study. We calculate three average scores: for in-domain test sets based on CNN/DM documents, for out-of-domain test sets based on XSum documents, and for the original datasets from TRUE.

**In-Domain Results** Most methods outperform the corresponding ANLI-only baseline, demonstrating the usefulness of synthetic data. Predictably, all methods improve with larger models and a complementary effect is often observed when mixing synthetic data with ANLI. The best results are obtained by mixing ANLI with Falsesum or TrueTeacher data and using T5-11B, with a substantial improvement over the corresponding ANLI-only baseline (in-domain score increase from 81.1 to 87.9).

**Out-of-domain Results** While most methods perform well in-domain, their performance drops significantly on the out-of-domain test sets. Most of the evaluated methods underperform the corresponding ANLI-only baseline with similar model capacity. For some methods, performance deteriorates dramatically; e.g. Falsesum – despite its impressive in-domain performance, its out-of-domain score falls significantly below the ANLI-only baseline. This suggests that some methods overfit to documents from the distribution used to generate the synthetic data. Based on this finding, we encourage future research to prioritize out-of-domain evaluation. Interestingly, even though TrueTeacher's relative improvement is smaller com-

pared to the in-domain setup, it is still the only method with higher out-of-domain score compared to the corresponding ANLI-only baseline. This demonstrates the robustness of TrueTeacher to domain shift, which may be due to the use of model-generated summaries that increase the variability of the resulting synthetic data.

**Overall Results on TRUE** Due to the poor out-of-domain performance of the existing methods, TrueTeacher is the only method that consistently outperforms the ANLI-only baseline on the TRUE benchmark. Notably, TrueTeacher + ANLI with T5-base (81.9) performs on par with the ANLI-only baseline using T5-11B (82.0). Additionally, the TrueTeacher-based variant using T5-11B (85.2) already performs on-par with the 540B LLM teacher (84.9, Table 2), even though we used only 100k synthetic examples in this experiment, and did not use ANLI data. When comparing TrueTeacher + ANLI with T5-11B and 100k examples (Table 3) to the equivalent variant using the full dataset (Table 2), we observe a performance increase (86.4 → 87.8), which demonstrates TrueTeacher's scalability. We conclude that TrueTeacher yields high quality data and generalizes well for new domains, which we attribute to the usage of model-generated summaries.

### 4.3 Qualitative Analysis

Figure 3 presents a case study with a randomly sampled document, and the corresponding *inconsistent* summaries generated with each of the evaluated

CNN/DailyMail ID: 372f7e02e5bb17bac3a1b2260c6ac78414f97ee3

Article: *LOS ANGELES, California (CNN) -- Los Angeles firefighters and city crews worked* for several hours *Tuesday to rescue one of their own: a 22-ton* firetruck *that was nearly swallowed by a* water-logged sinkhole. *Two firefighters crawled out of the truck's windows after it sank Tuesday morning. No one was injured. The incident happened after four firefighters took the truck to the San Fernando Valley neighborhood of Valley Village, where flooding had been reported… …*

Gold Summaries:
1. Los Angeles firetruck nearly swallowed by sinkhole Tuesday morning.
2. Firefighters in truck were responding to flooding call when incident happened.
3. Two firefighters escaped truck through windows; no injuries reported.

| FactEdit | Firefighters in truck were responding rescue when incident happened . |
|---|---|
| DocNLI | Los Angeles firetruck nearly destroyed by sinkhole Tuesday night . Firefighters in truck were responding to emergency call when it happened . Two firefighters escaped truck through windows ; no injuries reported . |
| FactCC | **LOS** LOS ANGELES, California ((CNN) - Los Angeles firefighters and crews worked Two on Tuesday to rescue one of their own own: a 22-ton fire engine nearly swallowed by a sinkhole filled with water**water**. |
| Falsesum | Los Angeles firetruck nearly swallowed by water. |
| TrueTeacher | A firefighter has rescued a truck that sank in Los Angeles, **causing extensive flooding**. |

Figure 3: A case study comparing factually inconsistent summaries of the same document generated using different methods. Content replacements are highlighted using the same color for the original and the replaced text. Added content is in bold red font.

methods. **FactEdit** used the second gold-summary and replaced *"to flooding call"* with *"rescue"*, introducing a grammatical error rather than a factual error, demonstrating the potential problems with using lower-beam completions as proxy for factual errors. **DocNLI** uses all the gold summaries concatenated. While replacing *"morning"* with *"night"* introduces a factual error, three other edits fail to introduce factual errors, demonstrating the limitations of using simple word/entity replacements. **FactCC** used the first sentence from the article and successfully introduced factual error by an entity swap from *"firetruck"* to *"fire engine"*. The paraphrase highlighted in green increases the abstractiveness, but the paraphrase in orange introduces a grammatical error that is less likely to be made by a strong summarization model. The noise injection used by FactCC (duplicating or removing random tokens) is colored in red, but its usefulness is questionable. **Falsesum** uses the first gold summary, and its perturbation model predicts the removal of *"Tuesday morning"* and the replacement of the *"sinkhole"* argument with *"water"*, failing to introduce a factual error, since the sinkhole is referred to as *"water-logged sinkhole"* in the article. Finally, **TrueTeacher** uses an abstractive summary generated by a real summarization model.

| Class | #Ex. | Precision | Recall | F1 |
|---|---|---|---|---|
| Consistent | 41 | 80.0 | 97.6 | 87.9 |
| Inconsistent | 59 | 98.0 | 83.1 | 89.9 |

Table 4: Human evaluation results.

It introduces a nuanced factual error by replacing *"Los Angeles firefighters"* with *A firefighter* and also by hallucinating new content (the text in bold red font). This case study further illustrates the challenges of perturbing texts to introduce factual inconsistencies and re-iterates the importance in using model-generated summaries.

### 4.4 Human Evaluation

To further assess the quality of the synthetic data produced by TrueTeacher, we perform human evaluation carried out by domain experts.[13] We evaluate 100 examples from our dataset,[14] using binary judgements based on the attribution definition from Rashkin et al. (2021). The labeling accuracy of the sampled examples from our data stands at 89%, which demonstrates its high quality. Table 4 further presents the precision, recall and F1 scores for the *consistent* and *inconsistent* classes. More details on the human evaluation are available in §A.8.

### 4.5 Ablating Summary Distribution and Label Correctness

There are two key differences between TrueTeacher and perturbation-based synthetic data generation methods: (1) the distribution of the summaries[15] and (2) the correctness of the generated labels.[16] Each of these differences may lead to the better quality of TrueTeacher w.r.t the baselines. To measure the impact of each difference, we isolate them in a controlled ablation study. We create 2 ablated variants, using Falsesum as a recent baseline method for synthetic data generation. The results are presented in Table 5.

**LabelAblation** is an ablation created by labeling the document-summary pairs from Falsesum's data using FLAN-PaLM 540B.[17] Comparing

---

[13] 10 NLP researchers, each with at least one year of experience in factual consistency evaluation.

[14] We randomly sampled 50 positively and 50 negatively labeled examples from our synthetic dataset.

[15] Model-generated vs. human-written perturbed.

[16] Both methods may yield wrong labels. Perturbations might not introduce inconsistencies, as seen in §4.3, while TrueTeacher can have errors due to LLM mislabeling.

[17] We used the same 100k examples as Falsesum + ANLI baseline, and the same LLM prompt as in TrueTeacher.

| Variant | Summary Distribution | Labeling Quality | T5-11B | T5-Base |
|---------|---------------------|------------------|--------|---------|
| Falsesum + ANLI | Human-written perturbed | Falsesum | 80.8 | 73.5 |
| TrueTeacher + ANLI | Model-generated | FLAN-PaLM 540B | 86.4 (**+6.9%**) | 81.9 (**+11.4%**) |
| LabelAblation | Human-written perturbed | FLAN-PaLM 540B | 85.3 (**+5.6%**) | 78.9 (**+7.3%**) |
| SummaryAblation | Model-generated | Falsesum (proxy) | 85.5 (**+5.8%**) | 79.1 (**+7.6%**) |

Table 5: Average ROC-AUC on TRUE for the ablated variants. Falsesum + ANLI and TrueTeacher + ANLI are copied from Table 3 for reference.

LabelAblation to Falsesum + ANLI allows us to examine the effect of using FLAN-PaLM labels instead of the original Falsesum labels, while controlling for the summaries distribution. LabelAblation outperforms Falsesum + ANLI by **5.6%**, which shows that performance gains can be obtained using summaries generated with existing synthetic data generation methods combined with second-stage improved labeling quality. However, TrueTeacher is substantially simpler and also results in better performance.

**SummaryAblation** is an ablation created by flipping labels on a random portion of TrueTeacher's data, such that the *expected* labeling accuracy is similar to Falsesum (More details in §A.9). Comparing SummaryAblation to Falsesum + ANLI allows us to examine the effect of changing the summary distribution from *human-written perturbed* to *model-generated*, while controlling for the labeling quality. SummaryAblation outperforms Falsesum + ANLI by **5.8%**, a similar improvement as observed for LabelAblation (5.6%). This demonstrates that label correctness and summary distribution have a similar effect on the performance, but they also have a complimentary effect as the best performance of 86.4 ROC-AUC is obtained only when they are combined together.

### 4.6 Abstractiveness Analysis

Advances in large scale pretraining (Devlin et al., 2019; Lewis et al., 2020) and the availability of relevant datasets (Narayan et al., 2018), enabled rapid progress in *abstractive* summarization, which better imitates the way humans summarize (Koh et al., 2023) and is also preferred by humans (Goyal et al., 2022). This motivates us to focus on generating *abstractive* synthetic summaries.

We compare the abstractiveness degree of different methods using the extractive fragment *coverage* and *density* measures from Grusky et al. (2018). Following Utama et al. (2022) we multiply these

|  | Coverage ↓ | Density ↓ | Combined ↓ |
|--|-----------|-----------|-----------|
| FactEdit | 0.86 | 2.92 | 2.67 |
| DocNLI | **0.85** | 15.66 | 15.20 |
| FactCC | 0.93 | 8.16 | 7.93 |
| Falsesum | 0.88 | 2.98 | 2.76 |
| TrueTeacher | 0.86 | **2.41** | **2.15** |

Table 6: Average abstractiveness scores (lower is better), measured on a random sample of 5k examples.

measures to obtain a *combined* score.[18] Table 6 presents the abstractiveness scores, and a density plot is available in the Appendix (Figure 5). We observe higher abstractiveness for model-based methods (FactEdit, Falsesum and TrueTeacher), suggesting that rule-based methods might be less useful with the recent shift towards abstractive summarization. TrueTeacher produces the most abstractive summaries with lowest *combined* score.

## 5 Multi-Lingual Data Generation for Factual Consistency Evaluation

Utilizing a multilingual LLM enables a straightforward application of TrueTeacher to multiple languages. This contrasts with recent approaches that rely on NLP components only available for high-resource languages, e.g., information extraction (Utama et al., 2022; Balachandran et al., 2022). In this section, we examine TrueTeacher's usefulness for multilingual factual consistency evaluation.

We first generate multilingual synthetic data using TrueTeacher. This time we train a single summarization model by fine tuning mT5-XXL (Xue et al., 2021) on XLSum (Hasan et al., 2021) and use it to summarize documents from WikiLingua (Ladhak et al., 2020), which we then label for consistency with our LLM. For the purposes of this experiment we focus on a subset of WikiLingua documents in 4 languages: English (en), French

---

[18]We provide additional technical details in §A.6.

| Training data | # Improved languages | Avg. ROC-AUC Per lang. | Per ex. |
|---|---|---|---|
| ANLI+XNLI | - | 73.3 | 71.6 |
| +TrueTeacher en | 32 / 45 | 75.7 | 73.8 |
| +TrueTeacher en,fe,es,ge | **35 / 45** | **77.2** | **75.3** |

Table 7: Multilingual results on the mFACE test set.

(fe), Spanish (es) and German (de).[19] After generating the dataset for these 4 languages, we sample 100k examples, by randomly sampling 25k in each language with balanced labels (as illustrated in Table 9 in the Appendix). For ablation, we also create an English-only variant, by randomly sampling 100k English examples with balanced labels.[20]

We use the resulted data to train multilingual consistency evaluation models and evaluate them on the mFace test set (Aharoni et al., 2022), containing 3150 examples in 45 languages. As a strong baseline we follow Aharoni et al. and fine-tune mT5-XXL (Xue et al., 2021) on the ANLI (Nie et al., 2020) and XNLI (Conneau et al., 2018) datasets. We then assess whether adding our synthetic data to the training set can improve this model.

Table 7 presents the results overview, full results in all 45 languages are available in Table 10 (Appendix). Adding English-only summarization-based synthetic data, already improves results on 32 out of 45 languages and increases the avg. ROC-AUC from 71.6 to 73.8. Yet, using the same amount of multi-lingual examples improved the performance even more, with avg. ROC AUC of 75.3. This demonstrates the added value in generating multi-lingual synthetic examples using TrueTeacher, laying the ground for future work.

## 6 Related Work

Previous work proposed methods for generating synthetic training data for factual consistency evaluation, by perturbing gold summaries (Yin et al., 2021; Kryscinski et al., 2020; Balachandran et al., 2022; Utama et al., 2022; Soleimani et al., 2023).[21] A key advantage of TrueTeacher, is the ability to leverage real model-generated summaries, leading to superior performance and robustness. The utility of model-generated outputs was also highlighted by Wu et al. (2023), who proposed a weakly super-

vised consistency evaluation model that leverages probabilistic labels derived from aggregated scores of other consistency evaluation models. Our work proposes a simpler solution, that is also inherently multilingual.

Another line of work for adapting NLI-based models for summarization, focuses on better processing of long texts, splitting the documents into sentences to create shorter premise-hypothesis pairs (Laban et al., 2022; Schuster et al., 2022).

Recent work attempts to assess LLMs' capability for evaluating generative tasks (Kocmi and Federmann, 2023; Wang et al., 2023; Liu et al., 2023). Luo et al. (2023) evaluated ChatGPT (OpenAI, 2022) specifically on the task of factual consistency evaluation in summarization. Yet, Aiyappa et al. (2023) argued that ChatGPT's "closed" nature risks data leakage (training-test contamination).[22] Chen et al. (2023) performed a study of LLMs as factual consistency evaluators, using a variety of prompting methods.

Previous work also attempted to distill knowledge from LLMs (West et al., 2022; Hsieh et al., 2023), as well as to leverage LLMs for data annotation (Wang et al., 2021; Ding et al., 2022), and synthetic data generation (Agrawal et al., 2022; Liu et al., 2022; Bitton et al., 2023). As far as we aware, our work is the first to leverage LLMs for data generation for factual consistency evaluation.

## 7 Conclusion

We introduced TrueTeacher, a simple and highly effective method for generating synthetic data for factual consistency evaluation. Instead of perturbation of human-written summaries like done in previous work, TrueTeacher leverages realistic model-generated summaries, which are annotated by prompting a large language model.

Using our method, we generate a large-scale synthetic dataset, which we are making publicly available. Our experimental results show that this dataset substantially enhances the performance of a state-of-the-art model. In our systematic study, we compare TrueTeacher to existing approaches and further demonstrate its effectiveness and robustness. Our study highlights the importance of out-of-domain evaluation, which we hope will be adopted in future work. Lastly, we show that TrueTeacher generalizes well to multilingual scenarios, presenting additional advantage over existing methods.

---

[19]They are the most prevalent languages in PaLM's pre-training data (Chowdhery et al., 2022)

[20]Also based on WikiLingua, generated with the same process like the 25k English subset of our multilingual dataset.

[21]We provide extensive review of these methods in §3.3.

[22]While FLAN's instruction fine-tuning data is public.

## 8 Limitations

**Noisy synthetic data** TrueTeacher relies on a LLM for labeling model generated summaries. This process may result in some frequency of noisy synthetic examples for which the label is incorrect. This can affect the overall quality of the student model that trains on this data. In our experiments we validated the quality of our synthetic data with human evaluation, however this should be re-examined when generating data for new domains. In addition, we experimented with different filtering approaches, but found that training on filtered data with higher labeling accuracy, did not improve the performance of the student model. We encourage future work to further examine such automatic filtering.

**Reliance on LLMs** In this work we use a 540B LLM to label 1.4M model generated summaries. This requires non-negligible resources that may not be available to the whole community. To mitigate this, we release our collected synthetic data and the corresponding model checkpoint. In addition, the decreasing inference cost of proprietary LLMs, and the availability of open-source LLMs (Touvron et al., 2023) can further assist.

**Effect of low-resource languages** Our multilingual experiments (§5) focus on a subset of WikiLingua documents in only 4 languages: English (en), French (fe), Spanish (es) and German (de), that are the most prevalent in our LLM's pre-training data. As can be seen in our full results (Table 9 in the Appendix), our multilingual data successfully improves low-resource languages as well. We did not fully explore the effect of adding additional languages to our synthetic data, especially low-resource ones. We believe that there is a trade-off between language coverage and labeling quality. i.e, while generating the synthetic data in low-resource languages will increase language coverage, it can lead to poor labeling quality by our LLM. We did not fully explore the exact sweet-spot for how many languages to include in our synthetically labeled training data, leaving this for future work.

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

## A Appendix

### A.1 FLAN-PaLM Prompt Design

To apply FLAN-PaLM for factual consistency evaluation, we experimented with zero-shot, few-shot and chain-of-thought prompting strategies, and various formats for each strategy. We chose the best performing strategy and format, based on the accuracy on a development set.[23] Table 8 presents the accuracy of each prompt type on the development set. We observed only minor performance differences, and thus we opted for the simplest solution that is the zero-shot prompt. While we cannot know the exact reasons for why few-shot and chain-of-thought did not improve performance, we can offer potential explanations. (1) Since the model was fine-tuned on NLI datasets, it is able to effectively generalize to factual consistency evaluation, making further demonstrations via few-shot prompting unnecessary in this case. (2) The performance with the zero-shot prompt is already notably high (89%, §4.4) and thus our particular LLM is less likely to benefit from chain-of-thought prompting. (3) It could be the case that only a few reasoning steps are needed to evaluate consistency in our particular setup and thus chain-of-thought is not necessarily better in this case.

Below, we describe our top-performing zero-shot, few-shot and chain-of-thought prompts.

**Zero-shot Prompt**  Since FLAN-PaLM was instruction fine-tuned on NLI, we designed our prompt to resemble an NLI prompt (e.g. using "premise" and "hypothesis" instead of "document" and "summary"). Our final prompt is as follows:

> *Premise: {document} **Hypothesis**: {summary} **Can the hypothesis be inferred from the premise? Answer using "Yes" or "No" only.***

**Few-shot Prompt**  We use two few-shot examples, one `"consistent"` and one `"inconsistent"`. We randomly sample these examples from the development set examples shorter than 200 words.[23] We limit ourselves to two short examples since summarization examples can include long documents, and thus few-shot may lead to too long context length. Our final prompt is as follows:

---
[23]For development set we use the FactCC dataset (Kryscinski et al., 2020) with 1,431 examples containing summaries of documents from CNN/DailyMail, manually annotated for factual correctness. Following (Utama et al., 2022), we merge the dev and test sets.

> *Premise: (CNN) Desperate migrants from Africa and the Middle East keep heading to Europe, with 978 rescued Friday in the Mediterranean Sea, the Italian Coast Guard said Saturday via Twitter. The migrants were picked up 30 miles off the coast of Libya, said European Parliament member Matteo Salvini, the leader of Italy's far-right Northern League. In the first three months of 2015, Italy registered more than 10,000 migrants arriving, the International Organization for Migration said, and about 2,000 were rescued at sea during the first weekend of April in the Channel of Sicily. Most migrants recorded this year come from countries in West Africa as well as Somalia and Syria, the IMO said. They use Libya as a country of transit. At least 480 migrants have died while crossing the Mediterranean since the beginning of the year, often because of bad weather and overcrowded vessels used by smugglers, the IMO said. Sometimes the captains and crews abandon the ships, leaving passengers to fend for themselves. At this time last year, there were fewer than 50 deaths reported, the IMO said. Most of the migrants are asylum seekers, victims of trafficking or violence, unaccompanied children and pregnant women.*
> *Hypothesis: the migrants were picked up 30 miles off the coast of libya.*
> *Can the hypothesis be inferred from the premise? Answer using "Yes" or "No" only.*
> *Answer: Yes*
>
> *Premise: (CNN) A nuclear submarine being repaired at a Russian shipyard has caught on fire, according to a law enforcement source speaking to Russia's state-run news agency ITAR-Tass. "The submarine is in a dry dock," Tass reports, citing the source, and there is no ammunition on board. "The rubber insulation between the submarine's light and pressure hull is on fire," Tass reported. Russia's RIA Novosti news agency says insulation caught on fire as welding work was being done on the submarine. Tass reported that the fire began on a sub in the Zvyozdochka shipyard in northwestern Russia. Zvyozdochka spokesman Yevgeny Gladyshev told the news agency that the sub had been undergoing repairs since November 2013. "Nuclear fuel from the sub's reactor has been unloaded," he reportedly said. "There are no armaments or chemically active, dangerous substances, fissionable materials on it," Gladyshev said to Tass. "The enterprise's personnel left the premises when the submarine caught fire, no one has been injured. The fire presents no threat to people and the shipyard."*
> *Hypothesis: "the rubber insulation between the submarine's light and pressure hull is on fire," russia's ria novosti news agency says.*
> *Can the hypothesis be inferred from the premise? Answer using "Yes" or "No" only.*
> *Answer: No*
>
> *Premise: {document}*
> *Hypothesis: {summary}*
> *Can the hypothesis be inferred from the premise? Answer using "Yes" or "No" only.*
> *Answer:*

**Chain-of-thought Prompt**  Following Kojima et al. (2022) we append `"Let's think step by step"` to the prompt to facilitate a step-by-step reasoning before predicting the label. Our final prompt is as follows:

> *Premise: {document}*
> *Hypothesis: {summary}*
> *Q: Can the hypothesis be inferred from the premise?*
> *Answer using "Yes" or "No" only.*
> *A: Let's think step by step*

This prompt successfully unlocked a step by step reasoning. Below is FLAN-PaLM's response format, where `{answer}` is either `"yes"` or `"no"`.

> *{reasoning steps}.* **So, the answer is {answer}.**

Example input:

> **Premise:** *(CNN) Georgia Southern University was in mourning Thursday after five nursing students were killed the day before in a multivehicle wreck near Savannah. Caitlin Baggett, Morgan Bass, Emily Clark, Abbie Deloach and Catherine (McKay) Pittman – all juniors – were killed in the Wednesday morning crash as they were traveling to a hospital in Savannah, according to the school website. Fellow nursing students Brittney McDaniel and Megan Richards were injured as was another person, who was not identified by the Georgia State Patrol. The young women were on their way to finish their first set of clinical rotations ... ... ...*
> **Hypothesis:** *georgia southern university was in mourning after five nursing students died.*
> **Q: Can the hypothesis be inferred from the premise?**
> **Answer using "Yes" or "No" only.**
> **A: Let's think step by step**

The output for this example is:

> *Georgia Southern University was in mourning Thursday after five nursing students were killed the day before in a multivehicle wreck near Savannah.* **So, the answer is yes.**

## A.2 Inference with FLAN-PaLM

We used the zero-shot prompt (see §A.1). The vast majority of FLAN-PaLM's responses were either `"Yes"` or `"No"`, and a tiny fraction of the responses were `"It's impossible to say"`.

During the labeling phase, we let FLAN-PaLM generate the output (predict mode), and label as `"consistent"` if the generated output is `"Yes"` and `"inconsistent"` in case the output is `"No"`. We discard the `"It's impossible to say"` examples. In order to measure ROC-AUC in a binary classification setting, we compute the model's probability of generating `"Yes"` (score mode) and use it as the example-level factual consistency score.

## A.3 Fine tuning T5

We fine tune our T5 models for factual consistency evaluation using the following input format:

| Prompt type | Dev accuracy |
|---|---|
| zero-shot | 93.6 |
| few-shot | 93.2 |
| chain-of-thought | 93.8 |

Table 8: FLAN-PaLM accuracy on the development set[23] using different prompting strategies.

| Language | ISO 639-1 | consistent | inconsistent |
|---|---|---|---|
| English | en | 12,500 | 12,500 |
| Spanish | es | 12,500 | 12,500 |
| French | fr | 12,500 | 12,500 |
| German | de | 12,500 | 12,500 |
| total | | 50,000 | 50,000 |

Table 9: Our multilingual dataset statistics.

> **Premise**: *{document}* **Hypothesis**: *{summary}*

The model is trained to predict `"1"` if the summary is factually consistent and `"0"` otherwise. We use a learning rate of $10^{-4}$ and a batch size of 32. During training, we use a maximum input length of 512 tokens and truncate the premise if needed.[24] During inference we use a maximum input length of 2048 tokens. We train for a maximum of 20 epochs, evaluate a checkpoint every 1k steps and choose the checkpoint with the best ROC-AUC on a development set.[23] In our study we make sure to use the same training regime for all baselines.

The ANLI-only results in Table 3 are from our experiments, while in Table 2 we use the results reported in previous work.

For the summarization models we fine tune the corresponding T5 models on the XSum training set (Narayan et al., 2018) in a similar fashion and use the ROUGE score on the XSum development set as a stopping criteria.

## A.4 Additional Details About Our Dataset

As mentioned in §3.1, we create the dataset based on documents from CNN/DailyMail (Hermann et al., 2015). We do not use the gold summaries, and we only use examples from the training set.

In our experiments with the full dataset (§4.1), we balance the labels by randomly sampling 475,563 positive examples (see Table 1).

---

[24]In early experiments we saw that training with longer maximum input length resulted with comparable performance.

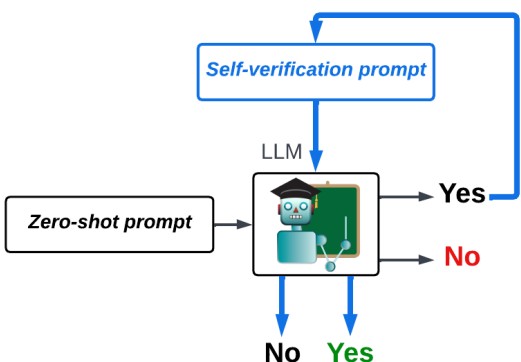

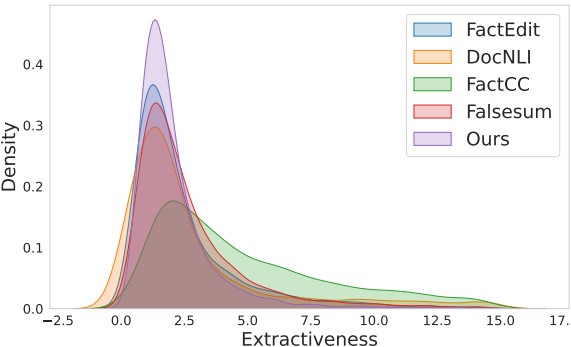

Figure 4: Self-verification prompting. If the LLM classified the summary as consistent, we prompt it again and ask it for its certainty. If the answer is "Yes" (consistent with the original reasoning), we keep the example, otherwise we filter it out.

## A.5 Data Filtering with Self-verification

As mentioned in §3 we also explored data filtering based on prompting FLAN-PaLM for self-verification. Our proccess is based on 3 steps. (1) Detect potential examples in our dataset that are likely to be labeled incorrectly by the LLM. (2) Prompt the LLM to self-verify its earlier prediction and filter out examples that the model is uncertain of. This leads to a smaller dataset with improved labeling accuracy. (3) Train the factual consistency evaluation model on the filtered dataset. This approach is based on 2 observations:

1. In early experiments, we saw that our LLM has extremely high precision for the inconsistent class. This can also be seen in our human evaluation (Table 4). This means that almost all the errors occur when the LLM predicts that the summary is consistent. Following this, we only consider filtering examples classified as consistent by the LLM.

2. Inspired by the work of Weng et al. (2023) and Madaan et al. (2023), we use a self verification prompt. If the LLM classified the summary as consistent, we prompt it again and ask it for its certainty. If the answer is "Yes" (i.e. it is consistent with the original reasoning path), we keep the example, otherwise we filter it out. This proccess is illustrated in Figure 4.

The self-verification prompt is as follows:

> *Premise: {document} Hypothesis: {summary} Are you sure that the summary can be inferred from the document? Answer using "Yes" or "No" only.*

This approach filtered-out 15% of the dataset.

Figure 5: Visualization of the density of the combined abstractivness score. The plot is actually measuring the extractiveness degree, so lower x-values mean higher abstractiveness.

When we qualitatively analyzed the filtered examples, it seems that the majority of the filtered examples indeed had a wrong label, and that applying this filtering mechanism increases the labeling accuracy by approximately 5%.

While this filtering mechanism results in higher labeling accuracy, we did not observe a performance gain when filtering the training data in this way. For TrueTeacher + ANLI with T5-11B (on a sample of 100k examples) we got an average of 86 ROC-AUC on TRUE using the filtered data, slightly below the 86.4 using the unfiltered data (Table 3). As mentioned in Footnote 9, we attribute this to the fact that the labeling accuracy is high to begin with (89%, section 4.4) and that the model is likely robust to some amount of labeling noise. Following this, for simplicity, our official method does not use filtering.

## A.6 Abstractiveness Analysis: Additional Details

As our backbone metrics we use the Extractive Fragment *Coverage* and *Density* measures defined by Grusky et al. (2018). *Coverage* measures the percentage of words in the summary that are part of an extractive fragment with the article, quantifying the extent to which a summary is derivative of a text. *Density* measures the average length of the extractive fragment to which each word in the summary belongs, quantifying how well the word sequence of a summary can be described as a series of extractions. Our *Combined* score is obtained by multiplyng the *Coverage* and the *Density* scores, similar to Utama et al. (2022). To further illustrated the differences in the abstractiveness of different methods, we include a visualization of the density of the combined abstractivness score in Figure 5.

|  | ANLI+XNLI | +100K en | +100K en/es/de/fe |
|---|---|---|---|
| amharic | 63.1 | 67.2 | **68.6** |
| arabic | 87.8 | **89.0** | 87.7 |
| azerbaijani | 59.6 | **68.6** | 65.5 |
| bengali | 90.4 | 94.3 | **98.5** |
| burmese | 59.0 | **64.5** | 57.9 |
| chinesesimp. | 87.6 | 86.4 | **89.9** |
| chinese trad. | 82.5 | 82.6 | **83.2** |
| english | **80.2** | 74.7 | 80.0 |
| french | 91.9 | 94.1 | **97.1** |
| gujarati | 50.8 | **52.0** | 51.5 |
| hausa | 69.5 | 67.7 | **73.7** |
| hindi | 72.2 | 79.9 | **86.5** |
| igbo | 62.2 | 62.8 | **75.7** |
| indonesian | 77.6 | 84.1 | **85.8** |
| japanese | 97.7 | 98.9 | **99.6** |
| kirundi | 83.5 | 89.3 | **90.4** |
| korean | 87.3 | 82.3 | **89.9** |
| kyrgyz | 70.1 | 77.4 | **79.0** |
| marathi | 75.2 | **78.7** | 73.6 |
| nepali | 55.2 | **59.1** | 57.2 |
| oromo | 81.2 | **83.7** | 83.3 |
| pashto | 56.4 | **68.2** | 67.7 |
| persian | 43.5 | 42.3 | **45.8** |
| pidgin | 70.0 | **81.4** | 77.1 |
| portuguese | **79.6** | 79.5 | 79.0 |
| punjabi | 77.7 | **81.5** | 78.2 |
| russian | **88.8** | 85.1 | 81.2 |
| scottish gaelic | 59.0 | 58.8 | **63.1** |
| serbian cyrillic | 84.2 | 79.3 | **85.5** |
| serbian latin | 39.7 | 42.2 | **43.6** |
| sinhala | 72.9 | 74.9 | **76.1** |
| somali | 85.1 | **88.6** | 86.6 |
| spanish | 80.7 | 85.9 | **89.1** |
| swahili | 88.1 | 89.2 | **92.2** |
| tamil | 63.9 | **69.8** | 66.0 |
| telugu | 55.9 | **62.3** | 60.4 |
| thai | 78.8 | 83.8 | **86.8** |
| tigrinya | 79.9 | 82.9 | **86.1** |
| turkish | **87.0** | 86.6 | 86.6 |
| ukrainian | 55.5 | **67.0** | 65.9 |
| urdu | 69.0 | 63.8 | **75.3** |
| uzbek | 54.6 | **59.3** | 58.8 |
| vietnamese | **89.8** | 84.4 | 88.1 |
| welsh | 83.0 | 83.4 | **83.9** |
| yoruba | 69.0 | 69.0 | **77.2** |
| # wins | 5 | 15 | **25** |
| # > ANLI+XNLI | - | 32 | **35** |
| Per lang. avg. | 73.3 | 75.7 | **77.2** |
| Per example avg. | 71.6 | 73.8 | **75.3** |

Table 10: ROC-AUC results on the mFace test set.

## A.7 Using the mFace dataset

In §5 we report results on the mFace dataset (Aharoni et al., 2022). Aharoni et al. performed large scale human evaluation of summaries of documents from the XLSum corpus (Hasan et al., 2021), produced by different summarization models. Each summary was rated for quality, attribution and informativeness. We use the attribution scores in our work. The attribution evaluation is based on the attribution definition provided in Rashkin et al. (2021), with the participants asked "*Is all the information in the summary fully attributable to the article?*". In our work we use the average attribution score (between 0 to 1) and treat summaries as factually consistent if the score is larger than 0.5. We focus on the test split of XLSum containing 3150 examples in 45 languages (i.e., 70 examples in each language). In §5 we refer to Table 7 with the results overview, and we provide the full results for all languages in Table 10.

## A.8 Human Evaluation

We instructed the participants to review the document and its corresponding summary, and to evaluate the summary based on the attribution definition provided by Rashkin et al. (2021), using binary judgements. To avoid a common confusion between factual inconsistency and contradiction, we also provided the following instruction:

> In this task you will evaluate the factual consistency of a system-generated summary. The system's goal is to summarize the original source document, while remaining truthful to it. Your goal is to evaluate whether the system-generated summary is consistent w.r.t. the source document. Summary will be considered consistent if all of the information in the summary can be verified from the source document (i.e., for the summary to be inconsistent, the document does not necessarily need to contradict it, it can also fail to support some facts).

In an early experiment, we found that using crowd workers without domain expertise and substantial time investments resulted in extremely low-quality ratings. Following this, all our raters were NLP researchers, each with at least one year of specific experience in the task of factual consistency evaluation, with significant time allocation and no more than 10 examples per rater.[25] These steps ensured high quality ratings.

## A.9 Adding noise to TrueTeacher

In §4.5 we create SummaryAblation by flipping labels to a random portion of TrueTeacher's data, such that the expected labeling accuracy is similar to Falsesum. Falsesum's labeling method is coupled with the data generation, thus we need an approximation for its labeling quality. We estimate Falesum's labeling accuracy as 83.5%, according to Utama et al. (2022)'s human evaluation (we average the Intrinsic and Extrinsic results), while ours is 89% (§4.4). So to mimic Falsesum's quality we flipped TrueTeacher's labels in order to add additional 5.5% errors.

---

[25]We found that it is sufficient to use one rater per example (unlike in our experiments with the crowd workers).