# OpenReview forum: "TrueTeacher: Learning Factual Consistency Evaluation with Large Language Models"
_EMNLP/2023/Conference — EMNLP 2023 Main_

### Official Review · Reviewer_TpWY · 2023-07-22

**Soundness:** 4

**Excitement:**

4: Strong: This paper deepens the understanding of some phenomenon or lowers the barriers to an existing research direction.

**Paper Topic And Main Contributions:**

The paper introduces TrueTeacher, a method to generate large-scale training data for improving factual consistency evaluation in summarization.

__Method:__ TrueTeacher first use different summarization models to produce large-scale machine-generated summaries, then use a 540B large language model to annotate the factual consistency of the machine-generated summaries. TrueTeacher can be considered as a distillation method (540B LLM -> 11B T5 model).

__Empirical Findings:__
* Models trained with data generated by TrueTeacher (combined with ANLI data) achieves state-of-the-art results on the TRUE benchmark.
* With a controlled analysis in 4.3 on different synthetic data generation method, the authors demonstrates the effectiveness of TrueTeacher compared to other methods such as FactEdit, DocNLI, FactCC, FalseSum; moreover, TrueTeacher outperforms these methods on out-of-domain evaluation.
* An additional set of experiment is done in a multilingual setting, showing that TrueTeacher is also applicable and brings improvements in such settings.

**Questions For The Authors:**

Line 335: I find the claim on TrueTeacher+ANLI outperforming 540B LLM less convincing.
* 540B LLM is fully zero-shot and to my understanding does not involve any optimization at all (e.g., post-hoc calibration, prompt selection). On the other hand, TrueTeacher+ANLI has access to a development set (Line 1004). It would be fairer if the 540B LLM approach also utilize the development set.
* It is mentioned in Line 216 that few-shot / chain-of-thought prompting is not effective in early phase of the study, it would be useful to describe these efforts in more details.

Sec 4.3: From the qualitative analysis I think there are two main difference between TrueTeacher and the other synthetic data generation methods: (1) the distribution of the summary (machine-generated vs. human-written + perturbation); (2) the correctness of the generated labels (e.g., falsesum failing in introducing factual error). Which aspect is more important? Is there a way to separate these two aspects in a controlled way? Would love to hear the authors thoughts on this.

**Reasons To Accept:**

* TrueTeacher is a simple, effective and strong method for improving factual consistency evaluation in summarization.
* Detailed comparison and ablation study demonstrating that TrueTeacher outperforms other generation methods, improves out-of-domain generalization, and remains effective for multilingual settings.
* The 1.4M generated data (to be released with the paper) may be useful to the research community.

**Reasons To Reject:**

* Limited technical contribution: The proposed method can be considered as a distillation recipe and similar approaches have been studied for other NLP problems (e.g., commonsense knowledge and reasoning). To this end the technical contribution of this paper is limited.

**Reproducibility:**

3: Could reproduce the results with some difficulty. The settings of parameters are underspecified or subjectively determined; the training/evaluation data are not widely available.

**Reviewer Confidence:**

4: Quite sure. I tried to check the important points carefully. It's unlikely, though conceivable, that I missed something that should affect my ratings.

**Typos Grammar Style And Presentation Improvements:**

* It would be helpful to provide more context in the abstract; currently it covers many aspects and is a bit confusing to a first-time reader.
* Line 569: specifically

---

> ### Author Rebuttal · Authors · 2023-08-28
>
> Thank you for the positive feedback. We are happy that you found our work strongly sound and our results insightful, especially as we believe that our ablation study is a key contribution of this work in addition to the proposed method. We also believe that our dataset will be a useful resource for future research.
>
> We will now provide a detailed response to each of your concerns and will do our best to carefully revise our work following your feedback. Regarding your second question about the importance of the different aspects (data distribution vs. synthetic label correctness), we performed additional detailed ablation experiments with insights we find interesting.
>
> ------------------------------------------------------------------
>
> # TrueTeacher can be considered a distillation recipe, thus the technical contribution is limited
>
> We acknowledge that TrueTeacher is closely related to knowledge distillation, a point we've noted in the paper (line 123 and line 337), and we recognize the technical simplicity of our method. However, we'd like to emphasize several aspects of our contributions which we find exciting and also distinguish our work from typical “distillation work”. We will revise our paper to emphasize these points.
> -  The focus of our work is clearly not on distillation in NLP, but rather on the task of factual consistency evaluation. Accordingly, the design choices we made are motivated by limitations of recent synthetic data generation approaches for factual consistency evaluation that we improve upon using a fundamentally different approach. Thus, we see our work as relevant for the summarization community.
> - Distillation requires a dataset for labeling, and we put significant efforts in generating one tailored to our use case from scratch. To achieve this, we (1) trained a diverse set of models to generate summaries, hypothesizing that the best proxy for real factual errors can be obtained from the outputs of actual summarization models, and (2) trained these summarization models on an abstractive dataset to produce summaries that more closely resemble those generated by modern summarization models.
> - Other than distillation, we make additional substantial contributions: Our study reveals previously unknown limitations of the existing synthetic data generation methods, and encourages future researchers to perform domain-aware evaluation. Our work provides useful insights about the applicability of LLMs for factual consistency evaluation, and about the feasibility of multilingual data generation (which our work is the first to explore in this context).
> - As mentioned in the paper and the responses, we experimented with several advanced prompting techniques, such self-verification and few-shot / chain-of-thought prompting (line 217 and line 231), yet the simplest solution worked best empirically. We believe that the technical simplicity of TrueTeacher is actually an advantage, as this makes our approach easy to adopt. To ensure that TrueTeacher can be applied easily, we further release our dataset and a strong model trained on it (T5-11B w. ANLI + TrueTeacher full, Table 2). We hope that it will provide a good starting point for future work in this important field.
>
> ------------------------------------------------------------------
>
> # The claim that TrueTeacher+ANLI outperforms 540B LLM is less convincing
>
> You raised 2 concerns in this context:
> 1. The 540B LLM is fully zero-shot.
> 2. It would be fairer if the 540B LLM approach also utilized the development set.
>
> Thank you for pointing these out. Regarding (1), we would like to clarify that our LLM was instruction fine-tuned, including on NLI datasets, and we are using a prompt that resembles the NLI task in those datasets. Thus, even though our inference is in a zero-shot format, our setting is not fully zero-shot. This can also explain why few-shot prompting does not improve performance (see results below), since our LLM is already familiar with the task. For (2) we would like to clarify that the LLM-based approach utilizes the development set, since we used it to choose the best performing prompt. We explored several zero-shot, few-shot and chain-of-thought prompts, and for each prompt type we experimented with different formats. For the few-shot approach, we also used few-shot examples from the development set. These details are indeed missing in the current manuscript, we will add them to the camera-ready version.
>
> In terms of how can `TrueTeacher+ANLI` outperform the `540B LLM`, we hypothesize that while the LLM is trained on many tasks, the smaller model should only master one specific task (factual consistency evaluation), which is easier (as we noted in line 336). Additionally, the smaller model is trained on target-domain data which can further improve its performance [(Gururangan et al. (2020))](https://aclanthology.org/2020.acl-main.740.pdf). We will try to better clarify this in the camera-ready version.
>
> ------------------------------------------------------------------
>
> # Details about our few-shot / chain-of-thought prompting efforts
>
> Following your request we provide more details about these efforts, and we will revise the paper to include these details (and results). As mentioned above, we experimented with different formats and chose the best performing prompt on the development set. The details of the zero-shot prompt are provided in the paper (line 970). Below, we describe our top-performing few-shot and chain-of-thought prompts. In addition, we add a table with the accuracy of each prompt type on the development set. The table shows that the differences are minor, so we opted for the simplest solution that is the zero-shot prompt. We attribute this to the fact that our particular LLM was already exposed to NLI datasets, and is able to effectively generalize to factual consistency evaluation.
>
> ### Accuracy of Different Prompt Types on the Development Set
>
> | Prompt type       | Dev accuracy |
> |-------------------|--------------|
> | zero-shot         | 93.6         |
> | few-shot          | 93.2         |
> | chain-of-thought  | 93.8         |
>
> ### Final few-shot prompt
> We use 2 few-shot examples, 1 positive (consistent) and 1 negative (inconsistent). We randomly sample these examples from all the examples from the development set which are shorter than 200 words. We limit ourselves to 2 examples and only to short examples since summarization examples can include long documents, and thus few-shot may lead to too long context length. Our final prompt is as follows:
>
> > ***Premise:*** *(CNN) Desperate migrants from Africa and the Middle East keep heading to Europe, with 978 rescued Friday in the Mediterranean Sea, the Italian Coast Guard said Saturday via Twitter. The migrants were picked up 30 miles off the coast of Libya, said European Parliament member Matteo Salvini, the leader of Italy's far-right Northern League. In the first three months of 2015, Italy registered more than 10,000 migrants arriving, the International Organization for Migration said, and about 2,000 were rescued at sea during the first weekend of April in the Channel of Sicily. Most migrants recorded this year come from countries in West Africa as well as Somalia and Syria, the IMO said. They use Libya as a country of transit. At least 480 migrants have died while crossing the Mediterranean since the beginning of the year, often because of bad weather and overcrowded vessels used by smugglers, the IMO said. Sometimes the captains and crews abandon the ships, leaving passengers to fend for themselves. At this time last year, there were fewer than 50 deaths reported, the IMO said. Most of the migrants are asylum seekers, victims of trafficking or violence, unaccompanied children and pregnant women.*
> >
> > ***Hypothesis:*** *the migrants were picked up 30 miles off the coast of libya.*
> >
> > ***Can the hypothesis be inferred from the premise? Answer using "Yes" or "No" only.***
> >
> > ***Answer:*** *Yes*
> >
> > &nbsp;
> >
> > ***Premise:*** *(CNN) A nuclear submarine being repaired at a Russian shipyard has caught on fire, according to a law enforcement source speaking to Russia's state-run news agency ITAR-Tass. "The submarine is in a dry dock," Tass reports, citing the source, and there is no ammunition on board. "The rubber insulation between the submarine's light and pressure hull is on fire," Tass reported. Russia's RIA Novosti news agency says insulation caught on fire as welding work was being done on the submarine. Tass reported that the fire began on a sub in the Zvyozdochka shipyard in northwestern Russia. Zvyozdochka spokesman Yevgeny Gladyshev told the news agency that the sub had been undergoing repairs since November 2013. "Nuclear fuel from the sub's reactor has been unloaded," he reportedly said. "There are no armaments or chemically active, dangerous substances, fissionable materials on it," Gladyshev said to Tass. "The enterprise's personnel left the premises when the submarine caught fire, no one has been injured. The fire presents no threat to people and the shipyard."*
> >
> > ***Hypothesis:*** *"the rubber insulation between the submarine's light and pressure hull is on fire," russia's ria novosti news agency says.*
> >
> > ***Can the hypothesis be inferred from the premise? Answer using "Yes" or "No" only.***
> >
> > ***Answer:*** *No*
> >
> > &nbsp;
> >
> > ***Premise:*** *{document}*
> >
> > ***Hypothesis:*** *{summary}*
> >
> > ***Can the hypothesis be inferred from the premise? Answer using "Yes" or "No" only.***
> >
> > ***Answer:***
>
> ### Final chain-of-thought prompt
>
> Following [Kojima et., al (2023)](https://arxiv.org/pdf/2205.11916.pdf) we append`Let's think step by step` to the prompt to facilitate a step-by-step reasoning before predicting the label. Our final prompt is:
>
> > ***Premise:*** *{document}*
> >
> > ***Hypothesis:*** *{summary}*
> >
> > ***Q: Can the hypothesis be inferred from the premise? Answer using "Yes" or "No" only.***
> >
> > ***A: Let's think step by step***
>
>
> This prompt successfully unlocked step by step reasoning and the response format is of the form `{reasoning steps}. So, the answer is {answer}.`, where {answer} is either yes or no.
>
> Example input:
>
> > ***Premise:*** *(CNN) Georgia Southern University was in mourning Thursday after five nursing students were killed the day before in a multivehicle wreck near Savannah. Caitlyn Baggett, Morgan Bass, Emily Clark, Abbie Deloach and Catherine (McKay) Pittman -- all juniors -- were killed in the  Wednesday morning crash as they were traveling to a hospital in Savannah, according to the school website. Fellow nursing students Brittney McDaniel and Megan Richards were injured as was another person, who was not identified by the Georgia State Patrol. The young women were on their way to finish their first set of clinical rotations. "Today should have been a day of celebration for this bright group of students," at St. Joseph's/Candler hospital said in a Facebook posting. "It was their last day of clinical rotations ... in their first year of nursing school." Clinicals include hands-on instruction at a health care facility. A post commander for the Georgia State Patrol said a tractor-trailer smashed into an eastbound line of cars that had slowed for a prior accident on Interstate 16. "He came along from behind them and he just did not stop for those cars," Sgt. Chris Nease said. There were four passenger vehicles and three tractor-trailers involved in the 5:45 a.m. accident. The women who were killed were in two cars, a Toyota Corolla and a Ford Escape. One of their vehicles caught on fire, Nease said, but it will take an investigation to determine whether the women died on impact. CNN Savannah affiliate WTOC reported one witness tried to help. "Right about the time I got here, the car was just about catching on fire," Cayne Monroe told the station. "The car just burned up really quickly. And I run up there, but there was nothing anyone could do. I've never witnessed something like that in my life. It was pretty tragic." The state patrol said the truck driver is from Louisiana. The 55-year-old man had not been charged as of Thursday evening, Nease told CNN. "Every one of our students contributes in no small measure to the Eagle Nation," university President Brooks A. Keel said in a statement. "The loss of any student, especially in a tragic way, is particularly painful. Losing five students is almost incomprehensible." Georgia Southern flew flags at half-staff and counseling was offered to students. A campuswide vigil was held Thursday night. On the university's Twitter page, a tear was added to the profile logo of the eagle mascot. The school has a student body of about 20,000 and is in Statesboro, about 60 miles from Savannah. "You could tell that they really loved what they did," Sherry Danello, vice president of patient care services and chief nursing officer at St. Joseph's/Candler, said on the hospital's Facebook posting. "They didn't just go through the task, they really connected to the patients." Luke Bryan, a country music star and school alumnus, tweeted his condolences: "Praying for everyone at Georgia Southern and the families who lost loved ones." CNN's Matthew Stucker contributed to this report.*
> >
> > ***Hypothesis:*** *georgia southern university was in mourning after five nursing students died.*
> >
> > ***Q: Can the hypothesis be inferred from the premise? Answer using "Yes" or "No" only.***
> >
> > ***A: Let's think step by step***
>
>
> The output for this example is:
>
> > *Georgia Southern University was in mourning Thursday after five nursing students were killed the day before in a multivehicle wreck near Savannah. So, the answer is yes.*
>
>
> ------------------------------------------------------------------
>
> #  The contribution of the data distribution vs. the correctness of the labels
>
> Thank you for this question. We believe the analysis below, which we conducted following your question, provides additional insights that will greatly improve our work. We performed an ablation study where we control the two aspects, i.e. the distribution of the summaries vs the quality of the generated labels. A TL;DR is that we show that they are both equally important, and have a complimentary effect when combined together. We describe our ablation study in detail below, which we will add to the paper. We will also revise the introduction to include both factors as motivation for TrueTeacher, stressing that perturbations of gold-summaries not only lead to unnatural summaries distribution, but they can also fail to introduce factuality errors.
>
> ### Our additional ablation study
>
> To control for the summaries distribution and the labeling quality, we created 2 ablated variants, using Falsesum as a recent relevant baseline method for synthetic data generation. Results on TRUE are presented in the table below. The first row (**variant #1**) is the `Falsesum + ANLI` variant from the paper (see Table 3), where the summaries distribution is **human-written + perturbation** and the labeling quality is of the **Falsesum** method. The last row (**variant #4**) is the `TrueTeacher + ANLI` variant from the paper (see Table 3), where the summaries distribution is **model-generated** and the labeling quality is of our **LLM**. Our ablated variants are in the second and the third rows (**variants #2 and #3**).
>
> | Variant | Summary Distribution                                 | Labeling Quality                      | T5-11B (ROC AUC) | T5-Base (ROC AUC)|
> |-----------|---------------------------------------------------------|-----------------------------------------|-------------------------|---------------------------|
> | 1          | Human-written + perturbation (Falsesum)  | Falsesum                                | 80.8                      | 73.5                         |
> | 2          | Human-written + perturbation (Falsesum)  | LLM                                         | 85.3 (**+5.6%**)  | 78.9  (**+7.3%**)     |
> | 3          | Model-generated (TrueTeacher)                 | Falsesum (proxy)                    | 85.5  (**+5.8%**) | 79.1  (**+7.6%**)    |
> | 4          | Model-generated (TrueTeacher)                 | LLM                                         | 86.4   (**+6.9%**) | 81.9   (**+11.4%**) |
>
> **Variant #2** was created by taking the document-summary pairs from Falsesum’s data and labeling them with our 540B LLM teacher. This variant has the same summary distribution as **variant #1** (i.e. **human-written + perturbation**) and the same labeling quality as **variant #4** (both are labeled using the LLM). Comparing **variant #2** to **variant #1** allows us to examine the effect of using our LLM for labeling (instead of using the original Falsesum labels), while controlling the summaries distribution. **Variant #2** improves **variant #1** by **5.6%**, while **variant #4** improves by **6.9%** \[1\]. This demonstrates that performance gains can be obtained using summaries generated with existing synthetic data generation methods combined with second-stage improved labeling quality. However, our data generation method is substantially simpler and also results in better performance (since **variant #4** outperforms **variant #3**).
>
> Finally, we would like to estimate the effect of changing the summaries distribution from **human-written + perturbation** to **model-generated**, while controlling the labeling quality. **Variant #3** is created by flipping labels on a random portion of TrueTeacher’s data, such that the *expected* labeling accuracy is the same as Falsesum’s \[2\]. Comparing **variant #3** to **variant #1** allows us to examine the effect of using **model-generated** summaries (instead of **human-written + perturbation**), while controlling the labeling quality. **Variant #3** improves by 5.8%, a similar improvement compared to **variant #2** (5.6%). This demonstrates that the data distribution and the correctness of the labels have a roughly similar effect on the performance, but they also have a complimentary effect and the best performance of 86.4 can be obtained only when they are combined together.
>
> \[1\]: We focus the analysis on T5-11B, yet we observe similar trends with T5-Base.
>
> \[2\]: Falsesum’s labeling method is coupled with the data generation, thus we need an approximation for its labeling quality. Falesum’s estimated labeling accuracy is 83.5% (according to [Utama et al. (2022)](https://aclanthology.org/2022.naacl-main.199.pdf)’s human evaluation), while ours is 89% (line 471). So to mimic Falsesum’s quality we flipped TrueTeacher’s labels in order to add additional 5.5% errors.

---

### Official Review · Reviewer_LjUF · 2023-08-02

**Soundness:** 4

**Excitement:**

3: Ambivalent: It has merits (e.g., it reports state-of-the-art results, the idea is nice), but there are key weaknesses (e.g., it describes incremental work), and it can significantly benefit from another round of revision. However, I won't object to accepting it if my co-reviewers champion it.

**Missing References:**



**Paper Topic And Main Contributions:**

This work is about a new framework TrueTeacher for summarization consistency by using pretrained language models to generate candidate summaries from documents without using the golden summaries and using another pretrained language model to label the consistency. TrueTeacher demonstrate competitive performance compared to previous single summarization models and the labeling mechanism is better combined with the summarization generation component in TrueTeacher compared to other summarization generation alternatively. Lastly, TrueTeacher has better generalizability in the multilingual settings.

**Reasons To Accept:**

- The framework incorporates the current progress of pretrained language models/large language models in factual consistency of summarizations.
- The experimental results are rich to demonstrate the performance of TrueTeacher under different scenarios, the finding of generalization ability of TrueTeacher could especially benefits the future research.

**Reasons To Reject:**

- The design of the labeling prompt mechanism is relatively simple, and still in a NLI style, would is likely to inherit the previous limitations of using NLI in factual consistency evaluation. The work does not fully address this problem or provide useful breakdown analysis of how it could be resolved.

**Reproducibility:**

N/A: Doesn't apply, since the paper does not include empirical results.

**Reviewer Confidence:**

3: Pretty sure, but there's a chance I missed something. Although I have a good feel for this area in general, I did not carefully check the paper's details, e.g., the math, experimental design, or novelty.

---

> ### Author Rebuttal · Authors · 2023-08-28
>
> Thank you for the positive feedback. We are happy that you found our work strongly sound and our results insightful.
>
> We will now provide a detailed response to each of your concerns/questions and will do our best to incorporate your feedback into the camera-ready version.
>
> ------------------------------------------------------------------
>
> # The prompt mechanism is relatively simple
> We experimented with several more complex prompting techniques such as **few-shot** and **chain-of-thought** prompting, as well as selecting examples with high confidence using a **self-verification** prompt (i.e. data filtering). For more details we refer you to:
> - Our response to **reviewer TpWY**, where we give extensive details about our **few-shot** and **chain-of-thought** efforts, since they specifically requested these details.
> - Our response to **reviewer AJ8x**, where we give extensive details about our data filtering effort (using a **self-verification** prompt), since they specifically requested these details.
>
> Eventually, the technically-simplest solution worked best, and thus we included it in our main results. While we believe that the simplicity of TrueTeacher will make it easy to adopt, we will also include the more complex approaches in the camera-ready version.
>
> ------------------------------------------------------------------
>
> # NLI-style prompt can cause inheritance of limitations of NLI models
>
> The NLI-style prompt performed well since the LLM was exposed to NLI tasks during instruction fine-tuning (lines 213-214). We believe that prompting an LLM with an NLI-style prompt is not equivalent to using an NLI model, as LLMs are pre-trained on a plethora of text and multiple NLP tasks and thus generalize well (e.g. [Wei et al. (2022)](https://arxiv.org/pdf/2109.01652.pdf)). Therefore, an LLM with an NLI-style prompt is more likely to generalize to the summarization domain (e.g. to long premises) compared to a model that was fine-tuned only on an NLI dataset. We also see that our LLM outperforms the SOTA NLI-specific model (84.9 vs. 82.7 ROC-AUC, Table 2). Furthermore,  our final model also benefits from training on a large amount of documents and model-generated summaries, which leads to superior performance, even compared to our LLM (87.8 vs. 84.9 ROC-AUC, Table 2). This is in line with previous work that demonstrated the utility of target-domain unlabeled data (e.g. [Gururangan et al. (2020)](https://aclanthology.org/2020.acl-main.740.pdf)). We will address this in the camera-ready version to provide more clarity.

---

### Official Review · Reviewer_AJ8x · 2023-08-04

**Typos Grammar Style And Presentation Improvements:** Foot note 2 should be in the page 3 f…
**Soundness:** 4

**Excitement:**

3: Ambivalent: It has merits (e.g., it reports state-of-the-art results, the idea is nice), but there are key weaknesses (e.g., it describes incremental work), and it can significantly benefit from another round of revision. However, I won't object to accepting it if my co-reviewers champion it.

**Missing References:**

N/A

**Paper Topic And Main Contributions:**

Factual consistency evaluation is often conducted using NLI datasets which shows limitations in evaluating summaries. Previous works have improved such models using synthetic data which is different from real-world data. LLMs could be an alternative solution to evaluating factual information while it is computationally expensive. To overcome this, the author proposed a method called TrueTeacher which generates synthetic data using LLMs. The author first trains summarization models and feeds documents to these models. Then the author makes LLMs to predict factual consistency labels and used further for factual consistency in summarization task which achieves the competitive performance compared to strong baselines.



**Questions For The Authors:**

Q1. What kinds of filtering mechanisms on LLMs labeled synthetic data have you tried that mentioned in line 232? What kind of samples have been filtered in which settings? Could you elaborate more on what your observations are?

Q2. In line 214, the authors mentioned they used closely-related NLI tasks for teacher model instruction fine-tuning. Is there a reason why you didn’t use publicly opened NLI datasets in this setting?


**Reasons To Accept:**

The paper is well-written and well-organized. The ablation studies cover various research questions. The authors also provide human evaluation of model-generated summaries that are classified by LLMs that indeed performs well on a factual inconsistency task.


**Reasons To Reject:**

The technological contribution of this paper is limited since the proposed method has been explored actively in the distillation setup.

**Reproducibility:**

3: Could reproduce the results with some difficulty. The settings of parameters are underspecified or subjectively determined; the training/evaluation data are not widely available.

**Reviewer Confidence:**

3: Pretty sure, but there's a chance I missed something. Although I have a good feel for this area in general, I did not carefully check the paper's details, e.g., the math, experimental design, or novelty.

---

> ### Author Rebuttal · Authors · 2023-08-28
>
> Thank you for the positive feedback. We are glad that you found our work strongly sound, well written and organized, and that you found our ablation studies insightful.
>
> We will now provide a detailed response to each of your concerns/questions, and will incorporate your feedback into the camera-ready version.
>
> ------------------------------------------------------------------
>
> # TrueTeacher can be considered a distillation recipe, thus the technical contribution is limited
>
> We acknowledge that TrueTeacher is closely related to knowledge distillation, a point we've noted in the paper (line 123 and line 337), and we recognize the technical simplicity of our method. However, we'd like to emphasize several aspects of our contributions which we find exciting and also distinguish our work from typical “distillation work”. We will revise our paper to emphasize these points.
> -  The focus of our work is clearly not on distillation in NLP, but rather on the task of factual consistency evaluation. Accordingly, the design choices we made are motivated by limitations of recent synthetic data generation approaches for factual consistency evaluation that we improve upon using a fundamentally different approach. Thus, we see our work as relevant for the summarization community.
> - Distillation requires a dataset for labeling, and we put significant efforts in generating one tailored to our use case from scratch. To achieve this, we (1) trained a diverse set of models to generate summaries, hypothesizing that the best proxy for real factual errors can be obtained from the outputs of actual summarization models, and (2) trained these summarization models on an abstractive dataset to produce summaries that more closely resemble those generated by modern summarization models.
> - Other than distillation, we make additional substantial contributions: Our study reveals previously unknown limitations of the existing synthetic data generation methods, and encourages future researchers to perform domain-aware evaluation. Our work provides useful insights about the applicability of LLMs for factual consistency evaluation, and about the feasibility of multilingual data generation (which our work is the first to explore in this context).
> - As mentioned in the paper and the responses, we experimented with several advanced prompting techniques, such self-verification and few-shot / chain-of-thought prompting (line 217 and line 231), yet the simplest solution worked best empirically. We believe that the technical simplicity of TrueTeacher is actually an advantage, as this makes our approach easy to adopt. To ensure that TrueTeacher can be applied easily, we further release our dataset and a strong model trained on it (T5-11B w. ANLI + TrueTeacher full, Table 2). We hope that it will provide a good starting point for future work in this important field.
>
>
> ------------------------------------------------------------------
>
> # Q1: What kind of filtering did you try?
> We will now describe our filtering attempts and will add the relevant details and results to the revised version.
>
> The main idea in our filtering experiment is:
> 1. Detect potential examples in our dataset that are likely to be labeled incorrectly by the LLM.
> 2. Prompt the LLM to self-verify its earlier prediction and filter out examples that the model is uncertain of. This leads to a smaller dataset with improved labeling accuracy.
> 3. Train the factual consistency evaluation model on the filtered dataset.
>
> Our filtering approach is based on 2 observations:
> 1. In early experiments, we noticed that our LLM has extremely high precision for the negative (inconsistent) class. This can be also seen in our human evaluation results in Table 4. This means that almost all the errors occur when the model predicts that the summary is consistent (failing to identify inconsistency). Following this, we only consider filtering examples that the LLM classified as consistent.
> 2. Inspired by the work of [Weng et al. (2023)](https://arxiv.org/pdf/2212.09561.pdf) and [Madaan et al. (2023)](https://arxiv.org/pdf/2303.17651.pdf), we use a self verification prompt. If the LLM classified the summary as consistent, we prompt it again and ask it for its certainty. If the answer is “Yes” (i.e. it is *consistent* with the original reasoning path), we keep the example, otherwise we filter it out.
>
> Our self verification prompt (the main prompt is presented in the paper in section A.1, line 973):
> > ***Premise:*** *{document}* ***Hypothesis:*** *{summary}* ***Are you sure that the hypothesis can be inferred from the premise? Answer using "Yes" or "No" only.***
>
> This approach filtered-out 15% of the dataset. When we qualitatively analyzed the filtered examples, it seems that the majority of the filtered examples indeed had a wrong label, and that applying this filtering mechanism increases the labeling accuracy by approximately 5%.
>
> While this filtering mechanism results in higher labeling accuracy, we did not observe a performance gain when filtering the training data in this way. For `TrueTeacher+ANLI` with `T5-11B` (on a sample of 100k examples) we got an average of 86 ROC-AUC on TRUE using the filtered data, slightly below the 86.4 using the unfiltered data (Table 3). As mentioned in footnote 7,we attribute this to the fact that the labeling accuracy is high to begin with (89%, line 471) and that the model is likely robust to some amount of labeling noise. Following this, for simplicity, we decided to remove this result from the paper. We will include it in the camera-ready version.
>
> ------------------------------------------------------------------
>
> # Q2: What kind of NLI tasks was the LLM trained on?
> To ensure anonymity, we defer the specification of the exact LLM we used to the camera-ready version. However, we would like to clarify that the LLM’s instruction fine-tuning dataset is fully disclosed and based only on **publicly available datasets** (including NLI datasets). The camera-ready version will provide the exact details for these datasets.

---

### Meta-Review · Area_Chair_uTBo · 2023-09-14

**Recommendation:** 4

**Metareview:**

The paper introduces TrueTeacher, a method to enhance factual consistency evaluation in summarization by generating extensive training data. TrueTeacher utilizes various summarization models to create machine-generated summaries on a large scale and employs a 540B large language model to assess the factual consistency of these summaries. This approach can be seen as a distillation method, with the 540B LLM as the teacher model for a smaller student model (T5). Models trained using TrueTeacher and ANLI data achieve state-of-the-art performance on the TRUE benchmark. The authors demonstrate its effectiveness through a controlled analysis comparing TrueTeacher to other methods such as FactEdit, DocNLI, FactCC, and FalseSum, especially in out-of-domain evaluations.
Additionally, TrueTeacher is proven to be applicable and beneficial in multilingual settings. The paper also promises to release 1.4M generated data for the research community's benefit. Even though the paper is well-written and includes comprehensive ablation studies and human evaluations of model-generated summaries, its technical contributions are limited.

---

### Decision · Program_Chairs · 2023-10-07

**Decision:**

Accept-Main

**Comment:**

The paper introduces TrueTeacher, a method to enhance factual consistency evaluation in summarization by generating extensive training data. TrueTeacher utilizes various summarization models to create machine-generated summaries on a large scale and employs a 540B large language model to assess the factual consistency of these summaries. This approach can be seen as a distillation method, with the 540B LLM as the teacher model for a smaller student model (T5). Models trained using TrueTeacher and ANLI data achieve state-of-the-art performance on the TRUE benchmark. The authors demonstrate its effectiveness through a controlled analysis comparing TrueTeacher to other methods such as FactEdit, DocNLI, FactCC, and FalseSum, especially in out-of-domain evaluations.
Additionally, TrueTeacher is proven to be applicable and beneficial in multilingual settings. The paper also promises to release 1.4M generated data for the research community's benefit. Even though the paper is well-written and includes comprehensive ablation studies and human evaluations of model-generated summaries, its technical contributions are limited.